# Candidate Genes and Their Expressions Involved in the Regulation of Milk and Meat Production and Quality in Goats (*Capra hircus*)

**DOI:** 10.3390/ani12080988

**Published:** 2022-04-11

**Authors:** Jose Ignacio Salgado Pardo, Juan Vicente Delgado Bermejo, Antonio González Ariza, José Manuel León Jurado, Carmen Marín Navas, Carlos Iglesias Pastrana, María del Amparo Martínez Martínez, Francisco Javier Navas González

**Affiliations:** 1Department of Genetics, Faculty of Veterinary Sciences, University of Córdoba, 14014 Córdoba, Spain; josalgadopardo@outlook.com (J.I.S.P.); juanviagr218@gmail.com (J.V.D.B.); angoarvet@outlook.es (A.G.A.); v32manac@uco.es (C.M.N.); ciglesiaspastrana@gmail.com (C.I.P.); ib2mamaa@uco.es (M.d.A.M.M.); 2Agropecuary Provincial Center of Córdoba, Provincial Council of Córdoba, 14014 Córdoba, Spain; jomalejur@yahoo.es; 3Institute of Agricultural Research and Training (IFAPA), Alameda del Obispo, 14004 Córdoba, Spain

**Keywords:** breeding, SNP, genomics, does, bucks, meat, milk

## Abstract

**Simple Summary:**

During the present decade, highly selected caprine farming has increased in popularity due to the hardiness and adaptability inherent to goats. Recent advances in genetics have enabled the improvement in goat selection efficiency. The present review explores how genetic technologies have been applied to the goat-farming sector in the last century. The main candidate genes related to economically relevant traits are reported. The major source of income in goat farming derives from the sale of milk and meat. Consequently, yield and quality must be specially considered. Meat-related traits were evaluated considering three functional groups (weight gain, carcass quality and fat profile). Milk traits were assessed in three additional functional groups (milk production, protein and fat content).

**Abstract:**

Despite their pivotal position as relevant sources for high-quality proteins in particularly hard environmental contexts, the domestic goat has not benefited from the advances made in genomics compared to other livestock species. Genetic analysis based on the study of candidate genes is considered an appropriate approach to elucidate the physiological mechanisms involved in the regulation of the expression of functional traits. This is especially relevant when such functional traits are linked to economic interest. The knowledge of candidate genes, their location on the goat genetic map and the specific phenotypic outcomes that may arise due to the regulation of their expression act as a catalyzer for the efficiency and accuracy of goat-breeding policies, which in turn translates into a greater competitiveness and sustainable profit for goats worldwide. To this aim, this review presents a chronological comprehensive analysis of caprine genetics and genomics through the evaluation of the available literature regarding the main candidate genes involved in meat and milk production and quality in the domestic goat. Additionally, this review aims to serve as a guide for future research, given that the assessment, determination and characterization of the genes associated with desirable phenotypes may provide information that may, in turn, enhance the implementation of goat-breeding programs in future and ensure their sustainability.

## 1. Introduction

Caprine farming has spread to almost every country in the world, due to the good prices and high value of goat-derived products (especially milk), attracting new farmers and investors [1]. The majority of the world caprine population is located in developing countries, occupying marginal territories under extreme climate conditions and held under elder farming systems [2]. This scene contrasts with that of Europe and North America, where otherwise, high-technological and intensive conditions rule the goat industry, which is highly focused on milk production and exploiting high-performance breeds subjected to genetic selection schemes [3]. In this context, the benefits that the domestic goat has obtained, as derived from the achievements made in the areas of genetics, nutrition and animal management, are rather limited in comparison to the level of integration that such techniques account for in other species.

The aforementioned framework evidences the secondary role to which caprine has been relegated within the scope of stockbreeding history [4]. This secondary position may be the result of two main conjoined facts; the traditional disregard of the caprine species as a destructive animal for pasture [5], and consumer preferences for other domestic species, which have conferred caprine-derived products with a low international market value, thus pushing caprine production to a marginal role in farming [6].

Recent archaeological findings indicate that the domestication of the goat took place more than 10,000 years ago in the ‘Fertile Crescent’ (Figure 1). This region is where the first settled agricultural communities of the Middle East and Mediterranean basin are thought to have originated, and would have covered the area from the Anatolian peninsula to the eastern territories of current Iran [7]. Wild bezoar (*Capra aegagrus*) and markhor (*Capra falconeri*) are thought to be the most likely ancestors of the domestic goat, according to phylogenetic studies implementing Y chromosome AMELY and ZFY sequences [8]. Additionally, other studies evaluating the major histocompatibility complex led to the possible inclusion of the Iberian mountain goat (*Capra pyrenaica*) and Alpine ibex (*Capra ibex*) in the history of goat domestication [3].

A higher tolerance to human handling and a better adaptability to the driven grazing may have been determining factors, which aimed to boost the popularity of certain animal populations [9]. Apart from the early breeding objectives that were sought following the domestication of the goat, the first civilizations became interested in the functional selection objectives linked to productive traits, such as the aptitude to captive breeding, prolificacy or body size [10].

The domestication and world dissemination of the species led to the first distinctive morphological traits of the original goat populations, such as the shape of the horns and ears [3]. This source of phenotypic variability could be the result of human/artificial selection, in addition to genetic drift and founder effects [11], which may explain the appearance of the characteristic traits in a lineage as a consequence of a narrow genetic base in its original population and its isolation. This may also be evidenced by other features, such as the presence of wattles, hair length or the wide variety of possible coat colours that have been developing as other distinctive traits in the first goat populations [3].

After centuries of their relationship with humans, natural selection for caprine adaptability to different environments and the artificial selection for productive, morphological and behaviour traits led to the appearance of 576 modern domestic goat breeds [10]. The Angora goat, whose presence in Phrygia and Cilicia (current Anatolia peninsula) was described 2400 years BC [12], was the first caprine breed in which a preliminary process of standardization was attempted. However, it would not be until 1890 and 1895 when the first caprine dairy goat breed standardization would take place in Switzerland, for the Saanen and Toggenburg goat breeds, respectively [13].

Although this event was a milestone and marked the beginning of caprine milk selection history [13], most of the significant advances would have to wait until the 1960s in France, when a bovine selection model would be applied to dairy goat production [14]. This turning point was promoted by the massive growth in the French cheese industry after the Second World War, which led to the development of intensive milking farm regions connected to the cheese factories. This improvement in animal production required farms to implement highly technical support, which brought about the routinization of progeny testing and artificial insemination [13]. It would take another decade for caprine meat breeding programmes (which have been dramatically scarcer than dairy goat breeding programs) to appear, with Boer goat breeding programmes being one of the few examples, appearing by the end of the 1970s in South Africa [15] (Figure 2).

The first documented register describing a caprine-selection-focused attempt dates back to 1962 [16], when the ‘Universidad de Puerto Rico’ studied the most profitable crosses between Puerto Rican local goat breeds and high-performance dairy goats. The study sought the most appropriate cross that would result in animals parallelly presenting the best adaptability, through the evaluation of goat kid survival rate, and the greatest productivity, through the evaluation of milk yield.

It was three years later (1965) when the first heritability estimations were performed in caprine [17]. This advance resulted from the integration of genealogical information as a compulsory step towards animal breeding estimations, which is a crucial step during the first stages of any breeding programme. The advances not only concerned the available genetic components or biostatistical tools, but also the revolution of phenotypic data collection. Contextually, the onset of 305-day lactation normalization in 1979 [18] made it possible to objectively compare the productivity of does from different breeds and controlled at different moments within lactation [19], which not only permitted farmers’ taking directed decisions based on factual data, but also laid the grounds for genetic evaluations. Still, the control of environmental effects was challenging, and reliable estimations were not feasible.

After the implementation of Best Linear Unbiased Prediction methods (BLUP) in caprine in the mid-1980s [20], along with the Animal Model [21] for breeding value calculations, the design of more complex selection schemes arose. This permitted the complete and reliable integration of genealogical information into genetic evaluations, but also the evaluation of animals that were phenotypically controlled in broadly distinct environmental conditions. This means that BLUP allowed for the separate estimation of the genetic and non-genetic (environmental) factors; hence, the heritable fraction of functional traits could be isolated and more appropriately controlled.

**Figure 2 animals-12-00988-f002:**
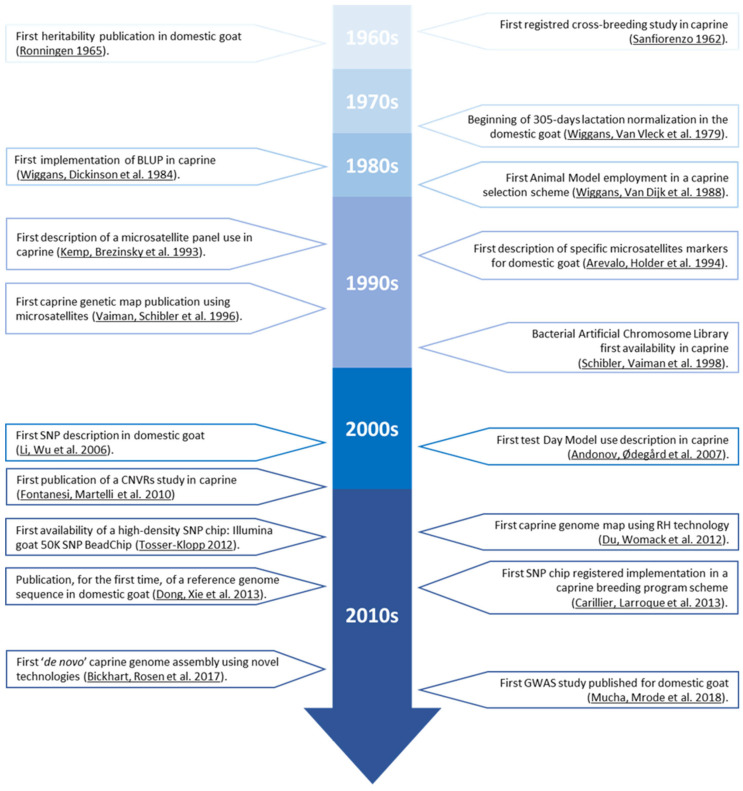
Timeline of caprine genetic advances milestones [16,17,18,20,21,22,23,24,25,26,27,28,29,30,31,32,33].

The ‘Genomic Era’ began with the discovery and utilization of microsatellite markers, which would be applied in the context of domestic goats as an extension of the use of bovine and ovine microsatellite panels in 1993 [22]. One year later, in 1994, the first description of caprine specific microsatellites was published [24]. The genomic information obtained from caprine microsatellite studies in these years permitted the development of studies based on the relationship between specific genomic regions, Quantitative Trait Loci (QTL), and desirable production traits [19].

QTLs are regions of the genome for which an association with the phenotypic variance of a certain trait has been proved [34]. Such an association may be supported by the fact that QTL regions may contain genes codifying for the specific regulation of the expression of a certain functional feature. For several years, many QTL were described using microsatellite genetic markers. However, even if they are still valid and preferrable when economic resources for research are scarce, the large size of some QTL [34] makes their mapping resolution and confidence intervals limited if the application of other, more efficient techniques is possible [35].

As previously mentioned, despite the fact that microsatellites offer a high degree of polymorphism for each marker, they are not as abundant as SNPs, and hence provide insufficient coverage of the genome [36]. In this regard, the first study of a caprine Single Nucleotide Polymorphism (SNP) was published in 2006 [37]. SNP would progressively replace microsatellites as the preferred genetic marker.

Additional milestones were quickly reached in the following years. For instance, the first study using the Canadian Test-Day Model in caprine was published in 2007 [38]. The Canadian Test-Day Model is a 12-trait random regression animal model, in which the traits are milk, fat, and protein test-day yields, and somatic cell scores on test days within each of the first three lactations. Test-day records from later lactations are not used. Random regressions (genetic and permanent environmental) are based on Wilmink’s three parameter function, which includes an intercept, regression on days in milk, and regression on an exponential function to the power –0.05 times days in milk (b0 + b1 × Exp(−0.05 × days in milk) + b2 × days in milk) [39]. This translates into genetic evaluations based on a better modelling of the lactation curve, providing more accurate results, which consequently enhances the selection progress.

These advances led to the publication of the first ‘Copy Number Variations Regions’ (CNVR) map for the domestic goat in 2010 [31]. Recent studies have shown that CNVR (intraspecific gains or losses of ≥1 kb of genomic DNA), represent important sources of variability of mammalian genomes (~0.4–25% of the genome). Their importance lies in the fact that CNVRs can change the gene structure and dosage, regulate gene expression and function and, hence, potentially have more effects than the most frequent single-nucleotide polymorphisms (SNPs) in determining phenotypic differences.

In 2012, the ‘International Goat Genome Consortium’ (IGGC) developed the first SNP chip for domestic goats; a high-density chip with 53,347 SNPs called ‘*Illumina 50K SNP BeadChip*’ (Illumina Inc., San Diego, CA, USA) [30]. In 2014, a new version of the high density SNP chip was developed, which is the most advanced goat SNP chip at present, called the ‘*Illumina 52K SNP BeadChip*’ a 60,000 SNP chip (Illumina Inc., San Diego, CA, USA), which is also carried out under the support of the IGGC [40]. Due to its robustness, low genotyping costs, automatic allele calling and the ability to interrogate the goat genome at high resolutions, these SNP chips were used to study the genetic diversity and population structure of native goats in various countries [41].

In 2013, after several attempts [23,25,42], the whole caprine genome was sequenced and optically mapped in China [29]. This advance made the reference genome sequence of the domestic goat available for the first time. The knowledge of the caprine genomic map was preceded by the first ‘Bacterial Artificial Chromosome Library’ published for caprine [43]. Recently, a new, ‘de novo’ genetic assembly (ARS1) has been developed. It allows for high-quality genomic mapping, thanks to the advanced sequencing methodology ‘*single-molecule PacBioRSII*’ [27]. This methodology uses single-molecule, real-time (SMRT) sequencing technology, which takes advantage of an immobilized DNA polymerase/template complex nested in thousands of small wells (called zero-mode waveguides). The value of these new technologies lies in the fact that they are characterized by their high-throughput characteristics. Hence, they provide the opportunity to produce millions of reads with an inexpensive sequencing.

The use of SNP analysis in goat association studies marked a milestone with the first ‘Genome Wide Association Study’ (GWAS) in 2018 [26]. Thanks to GWAS tools, researchers could obtain more in-depth knowledge of the association between specific genome sequences and the phenotypic variation in traits of interest. GWAS appeared as an alternative to QTL and microsatellite studies, given that it allows for the identification of narrower genomic regions [34,44].

These new genomic tools are promising in terms of genetic selection within breeding programmes, because they allow scientists to locate the candidate genes that are directly involved in phenotypic traits of economic impact. Hence, they enable the acceleration of genetic progress. Therefore, the present review provides a chronological analysis of the advances made in caprine genetics and genomics and describes the main candidate genes for caprine meat and milk production and quality that have been reported in the literature to date.

## 2. Data Collection

A comprehensive bibliographic analysis was performed of the documents published over the last 10 years. Reviews, articles, short notes, Doctoral theses, and Master’s theses were considered. The search for documents was performed using the ‘Google Scholar’ search engine (https://scholar.google.com/, accessed on 17 May 2021). This search engine was chosen as suggested by other papers developing bibliographic studies of a similar kind, as they comprise tools that enable data extraction for analysis [45,46,47]. The keywords chosen to perform the search of documents were “*caprine candidate genes, caprine meat genes, caprine milk genes*”. Semantic fields were also considered.

The selection criteria for the documents included in the final dataset comprised: (1) date, looking for those that were published during the past 10 years (2) content, giving priority to those based on the study of candidate genes, and (3) species, including those primarily dealing with the caprine species, but also considering those involving the caprine species, in addition to other species, such as the ovine. In the case of data overlapping in different documents, the data published in the most recent publication were chosen. Only two reviews based on caprine candidate genes were found. The first was published in 2009 and the second in 2020. Based on the bibliographic references that these reviews provided, it was possible to expand the information and to include more candidate genes involved in caprine meat and milk production. A remarkable lack of articles dealing with candidate genes in the caprine species was observed during the last two years of this study. Particularly, references exclusively dealing with the caprine species were sparse. Gene data (karyotype location and information) were obtained using the database of the National Centre of Biotechnology Information NBCI (https://www.ncbi.nlm.nih.gov/gene/). This database was last accessed on 5 July 2021 [48].

## 3. Candidate Genes Regulating the Expression of Economically Relevant Traits in Caprine

### 3.1. Candidate Genes Regulating the Expression of Goat Meat Breeding Criteria and Traits

Growth performance and weight gain are two of the most relevant traits in caprine farming, given that they directly relate to the shortening of puberty age and the number of kilograms of meat to be sold, hence benefitting the farmer [49]. Contextually, body weight increases until 5–6 years of age, at which time it begins to decrease. The literature indicates the existence of sexual dimorphism, with males being heavier than females and a progressive decrease in weight gain as the number of kids born increases, with single-birth kids being heavier than double-birth kids, and these, in turn, being heavier than multiple-birth kids. Among other important factors, the season of birth, nutrition management, farming system, and the age and birth order of the doe have been reported to influence body-weight-related traits [50].

The current selection framework is characterized by a lack of reciprocity between caprine dairy and meat production. Hence, in practice, this is one of the main challenges that the sector needs to face; improving dairy characteristics, which negatively correlate with those related to butchering production [51], while taking advantage of the sturdiness and improved adaptability of caprine breeds [52].

The breeding objectives used in artificial selection to obtain a desirable dairy morphology are the opposite to those seeking the enhancement of meat/butcher traits. This may explain the marked differences that exist between the characteristic shape of goat dairy breeds (thin and triangular forms), and meat breeds (short, rectangular and excellent body conformation) [49].

Carcass dressing percentage in goats is approximately 50–55%, and is usually lower than in ovine. This is due to the greater bone proportion in caprine, along with a different carcass fat deposition, which, in goats, is perivisceral rather than subcutaneous or intramuscular [53]. In the same way, the caprine carcass has a lower fat coverage; hence, moisture losses (which can reach 8%) are greater than those in sheep. However, the lower intramuscular fat deposition in caprine makes chevon meat a healthier alternative to lamb [54].

In terms of meat quality characteristics, organoleptic traits include muscle appearance, juiciness, texture, hardness, flavour and aroma [55]. Many factors condition the expression of these traits, including nutrition, exercise/physical activity, age, and method of slaughter and bleeding [55]. Regarding nutritional quality, chevon muscle is high in amino acid diversity and is very lean, presenting a lower degree of fat interspersion than other species (but with more polyunsaturated fatty acids) [6].

Body weight and butcher performance, body growth, body weight gain, kid survival rate, feed conversion index and dressing percentage of the carcass are among the most frequently considered quantitative traits within the framework of caprine selection. Table 1 presents a summary of the most frequently addressed breeding criteria and traits concerning selection for meat production in goats. Figure 3 presents a scheme of caprine candidate gene background regulating the expression of meat production.

#### 3.1.1. Actual and Potential Candidate Genes Regulating the Expression of Body Growth and Weight Gain

Among the genes for which a significant association has been reported with the regulation of the expression of body growth and weight gain, the following are the most relevant. First, the Adenylate Cyclase 1 gen (*ADCY1*), which encodes for a molecule involved in Cyclic Adenosine Monophosphate (cAMP) synthesis, is associated with chest width [60]. This gene regulates cellular division and mitosis [61], but also has been reported to hold regulatory control over Growth Hormone (GH) release [62].

Second, the Sorting Nexin 29 gene (*SNX29*), which has been described as a controller of the differentiation and proliferation of myocytes, has been directly ascribed to the complex of genes regulating the expression of muscle growth [63]. In parallel with this, its association with chest width and pin bone width, both relevant traits due to their association with dressing aptitude, has been recently reported [60]. Third, the LIM Domain Binding Factor 2 gene (*LDB2*), which encodes for a protein playing a crucial role in lymphocyte migration and atherosclerosis [64], has been shown to have a strong relationship with daily body weight gain in other species, such as poultry, as derived from GWAS studies [65].

Fourth, Myeloid-associated Differentiation Marker gene (*MYADM*) has been reported to be involved in the cytoplasmatic membrane synthesis of the myeloid-line cells and has impact on the erythrocyte morphology. Its association with post-weaning body weight in lambs has been described, which suggests it may be worth exploring in kids [66]. Fifth, the Insulin-like Growth Factor 1 Receptor gene (*IGF1R*) encodes for receptors for IGF-1 binding, modulating its blood concentration and activity, and has been shown to be associated with body growth and height in dogs [67].

Sixth, the Apolipoprotein L3 gene (*APOL3*) encodes for a protein involved in lipid transport and metabolism, and has been ascribed an interesting potential influence on milk and growth traits [68]. Seventh, and similar to the aforementioned, the Stromal Interaction Molecule 1 gene (*STIM1*) encodes for a membrane calcium transporter that is related to prolificacy traits [69], and has been suggested as a candidate gene for growth traits [70]. Eighth, the HMG-Box Containing 3 gene (*HMGXB3*), which has been described to be involved in cellular proliferation in neoplasia [71], has also been reported to be of great interest in meat production [70].

Ninth, among growth-hormone-linked genes, the Growth Hormone gene (*GH*), which encodes for the main hormone involved in the vertebrate corporal development, has been reported to be strongly related to body weight at birth and weight gain in goat kids [72]. In this regard, those genes linked to regulation or control of GH, such as the Growth Hormone Secretagogue Receptor gene (*GHSR*), may be relevant. For instance, *GHSR* encodes for a G protein-associated receptor that binds ghrelin, a GH-release-stimulating hormone [73]. Similarly, the Growth Hormone Receptor gene (*GHR*) encodes for the proteins to which *GH* binds, modulating the molecular mechanisms that this hormone sets in motion, which have a direct impact on corporal development [74].

Tenth, the insulin-like growth-factor-related genes, such as the insulin-like growth factor gene (*IGF-1*) play a pivotal role in the mammalian somatotrophic axis, participating in many metabolic functions involved in the regulation of the expression of growth, lactation, and reproduction traits. Indeed, its effects on the carcass conformation and fat distribution of cattle have been acknowledged in the research [75]. Additionally, the Insulin-like Growth Factor 2 Binding Protein 1 gene (*IGF2BP1*) encodes for a protein that takes part in the ‘*sonic hedgehog*’ route, which is a metabolic chain impacting body/organ growth and development [76]. This is also extendible to the Insulin-like Growth Factor Binding Protein 3 gene (*IGFBP3*), which encodes for a protein that regulates the activity of an IGF-1 subfamily, but is also associated with growth, body size and prolificacy traits [77]. Another ‘*sonic hedgehog*’ metabolic route participant is the Fibroblastic Growth Factor Receptor 1 (*FGFR1*), which unleashes a molecular-signals cascade associated with metabolism and organic development [78]. Eleventh, the Bone Morphogenic Protein gene (*BMP*) encodes for a transforming growth factor β (TGF-β) superfamily. Many of the genes in this complex, such as *BMP4* [79] and *BMP15* [80], have been reported to relate to growth traits.

Twelfth, the Nonsense-mediated mRNA Decay Factor gene (*SMG6*) has been suggested to affect growth due to its genome stabilization functions, given that genome instability inhibits mitosis and decreases cellular division and tissue growth [81]. Thirteenth, the Cell Adhesion Molecule 2 gene (*CADM2*) encodes for a synaptic signalling vector that is abundant in the brain and muscle, and has been described to be involved in weight at slaughter and corporal length [82]. Table 2 reports a summary of the candidate genes associated with body growth and weight gain that have been described in goat breeds, the chromosome location, exon counts, physiological function, and where to find them in the literature.

#### 3.1.2. Actual and Potential Candidate Genes Regulating the Expression of Body Size and Carcass Quality

The genes presented in this subsection have been reported to be associated with body size and carcass quality, either skeletal quality or muscle quality. First, the quest to find caprine specific genes related to the aforementioned traits is challenging, given the lack of existing species-specific literature. In this context, among the only examples found, the PR/SET Domain 6 gene (*PRDM6*) encodes for a histone methyltransferase, a molecule with an important role in corporal development and reproduction, and is, therefore, suspected to be related to body growth [88], and the Nerve Growth Factor gene (*NGF*) encodes for a crucial protein in the development and differentiation of the nervous system. However, it also plays an important role in muscle growth and prolificacy [89] in goats.

Interestingly, some of the genes for which an association with economically relevant traits has been reported have an additional relevance, as they coregulate the expression of dual-purpose traits; that is, they are associated with better carcass features while they regulate for better milk-quality traits. The Pituitary-specific Positive Transcription Factor 1 gene (*POU1F1 or PIT-1*) encodes for a positive regulator of many hormones, such as the Growth Hormone, Prolactin or Thyroid-stimulating Hormone, and has been shown to have a role in growth and lactation traits in caprine [90]. Furthermore, a certain degree of association is presumed with carcass conformation, owing to its relationship with body depth and leg size in bovine, alongside some milking morphological traits [91].

Additionally, regarding muscle development, the Leptin gene (*LEP*), which regulates the expression of a hormone that exerts its effects on daily voluntary food intake, energetic waste and corporal metabolic balance [92], has been proven to condition carcass weight, corporal fat yield and lean meat percentage in ovine [93]. The Myostatin gene (*MSTN*) is considered to be a highly relevant candidate gene for meat production, due to its widely studied association with body growth and muscle mass development in bovine [37].

Concerning bone quality and development, the T-box Transcription Factor 15 gene (*TBX15*) is highly expressed in mesenchymal precursor cells and chondrocytes, and has a direct impact on bone development [94]. This was also reported for the Drosha and DiGeorge Syndrome Critical Region gene 8 (*DGCR8*), which is involved in osteoclastic and bone-reabsorbing activity, and hence in the determination of skeletal development [95].

Regarding muscle development, the Cell Division Cycle 25 Homolog A gene (*CDC25A*) has been reported to take part in the cellular quiescence G1 phase and to be involved in myoblast differentiation in mice [96]. Additionally, the Solute Carrier Family 26 Member 2 gene (*SLC26A2*) encodes for a protein that seems to modify sulphate transporting residual activity, and is associated with short body size and skeletal dysplasia [97]. Table 3 summarizes the candidate genes associated with body size and carcass quality described in goat breeds to date, as well as their chromosome location, exon counts, physiological function and where to find them in the literature.

#### 3.1.3. Actual and Potential Candidate Genes Regulating the Expression of Carcass Lipidic Profile

Among the genes regulating the expression of the lipidic profile of caprine carcasses we first find the Peroxisome Proliferator-activated Receptors Gamma gene (*PPARγ*), which encodes for a protein that is directly related to carcass lipids, thanks to its regulatory activity over the expression of other genes involved in fat metabolism, promoting its mRNA synthesis and protein formation [99]. Second, the Lipoprotein Lipase gene (*LPL*), modulates fatty acid metabolism due to its implication in muscle lipid composition as reported in cattle [100].Third, following the aforementioned line, the Acetyl-Coenzyme A Carboxylase gene (*ACACA*) encodes for one of the most important enzymes involved in tissue fatty acid synthesis, and is associated with conformation traits and meat fat composition in bovine [101]. The Sterol Regulatory Element-binding Protein 1 (*SREBF1* or *SREBP-1c*) encodes for transcription factor family modulating lipid homeostasis [102] and seems to promote the expression of *ACACA* [103] and *PPARγ*’s transcription. Hence, its involvement in fatty acid metabolism is presumed [104]. Fourth, the Thyroid Hormone Responsive SPOT14 gene (*THRSP*) seems to mediate the expression of other genes that are involved in lipid synthesis, and is known to be related to the fatty acid profile in bovine muscle [105]. Fifth, the Carnitine Palmitoyl-transferase 1A gene (*CPT1A*), which encodes for a mitochondrial enzyme responsible for acyl-carnitine synthesis, has been proven to promote fatty acid transport into the mitochondria [106]. Certain genes are associated with fatty acid and adipose tissue metabolism in other species. For instance, a study in pigs [107] showed that the Retinol deshidrogenase-16 gene (*RHD16*) plays a role in the metabolic reactions in adipose tissue [107], while the Heart Fatty Acid-Binding Protein gene (*H-FABP*) is reported to be involved in lipid metabolism and seems to influence the intramuscular fat composition in pigs [108]. However, their role in caprine has yet to be confirmed [59]. Table 4 presents a summary of the candidate genes associated with carcass lipidic profile that have been described in goat breeds, their chromosome location, exon counts, physiological function and where to find them in the literature.

### 3.2. Candidate Genes Regulating the Expression of Goat Dairy Breeding Criteria and Traits

The international caprine milking sector comprised 215 million animals in 2019, which was one-quarter of the world caprine census [110]. Europe, with only 5% of the world caprine dairy census, produced 15% of the caprine worldwide milk production, which could mainly be ascribed to its high degree of specialization [1].

This high European productivity is supported on high-technology farms, on which goat breeds such as the Alpine, Saanen or Toggenburg have been bred under the scope of strict genetic selection schemes and breeding programmes [3]. However, this is only one side of the same coin and the opposite situation is found in developing countries. Specifically, in developing countries, genealogic control and productivity registers are barely available or are of poor quality [1]. The caprine European dairy sector is a well-regulated industry, where almost all the milk is processed into cheese [1]. Most of the farms’ income derives from milk production, while the sale of chevon for the meat industry is a marginal source of income, due to the relatively low value of caprine meat within developed countries [6].

According to García, et al. [111], milk quality is defined as the milk’s ability to tolerate the technological processes that lead to a market-demanded product in terms of nutritional value, food safety and sensorial parameters. This is especially relevant in the framework of the cheese industry, given that increasing milk protein and percentage (known as ‘dry’ or ‘cheese extract’) increases the chances to enhance farms’ rentability and profitability, even more than milk volume production (in litres) [112]. Furthermore, traditional breeding criteria (milk yield in volume; milk lactose, fat and protein percentages; fatty acid profile and omega 3 (technologic quality); and somatic cell count (hygienic quality) [111]) have recently been joined by postprocessing technological traits related to *cheese-making* quality indicators, in order to improve cheese production efficiency. Among these traits we find, for instance, milk cheese yield (%CY) or the dairy cheese efficiency (dCY) [112], and the newly implemented term of ‘recuperation’ (%REC) for each milk component (proteins and fat) from the junket.

#### 3.2.1. Actual and Potential Candidate Genes Regulating the Expression of Milk Production/Yield

Given the greater economic repercussion of goat dairy products against caprine-meat-derived products, it is understandable that the progress and knowledge on genes associated with milk yield and quality surpasses those for goat meat production-related genes. The evaluation of the literature suggested that a large fraction of the genes involved in meat quality- and/or yield-related traits correlatedly regulate the expression of milk yield- and content-related traits; hence, they indirectly correlate with the quality of milk and the products that derive from it, for example, cheese. For instance, the Pituitary-specific Positive Transcription Factor 1 gene (*POU1F1*) is associated with the expression of genes that encode for hormones regulating mammary gland development and milk production, such as prolactin, growth hormone, insulin-like growth factor 1, progesterone and oestrogens [113]. Among them, contextually, the Insulin-like Growth Factor 1 gene (*IGF-1*) is associated, as mentioned in the previous sections, with body growth and development or metabolism, but has also been reported to participate in the regulation of the expression of milk protein and fat [75], or the Vacuolar Protein Sorting 13 gene (*VPS13*) family, which is actively involved in milk production, as was proven in other mammalian domestic species [70]. For example, while the *VPS13C* gene seems to affect glucose homeostasis in high-productive milk cattle [114], the *VPS13B* gene was detected in a QTL related to leg length (which is associated with fertility, prolificacy and milk production) [115]. Among the genes which were found to be associated with dairy yield or quality, first, the Leptin gen (*LEP*) was reported as a candidate gene for milk yield and quality traits, due to its implications for the regulation of daily voluntary food intake, energetic distribution and metabolism in cattle [116]. Consequently, this brings about the implications of the Leptin Receptor gene (*LEPR*), which has also been reported to have an influence on milk traits, due to its function in leptin blood concentration and physiological activity [117]. Second, the ATP Binding Cassete Subfamily G Member 2 gen (*ABCG2*), besides its acknowledged membrane drug and xenobiotics’ transporting activity, is presumably linked to a decrease in milk production and an increase in milk protein and fat [118]. Third, the Growth Hormone Receptor gene (*GHR*) encodes for the GH transmembrane receptors that unleash the cellular mechanisms set off by this hormone, and is directly associated with carbohydrate and lipid metabolism, and involved in the beginning and maintenance of lactation [119]. Fourth, it is common for genes regulating the expression of milk yield to have a negative or positive correlation with milk composition (protein, fat, lactose and somatic cell count, among others). In this context, the Prolactin gene (PRL), which encodes for a physiological multi-function hormone, whose specific actions occur at the reproductive and lactation levels, and, hence, influences milk yield and milk protein and fat percentages [120]. In the same way, the Prolactin Receptor gene (PRLR), which encodes for a receptor associated with a wide range of endocrine functions, such as mammalian lactation and body growth, is one of the most frequently addressed candidate genes for milk traits in the literature, given its regulation of the expression of milk yield and lactose, protein and fat percentages [121]. Fifth, the high expression of these genes in the mammary gland seems to be responsible for their implications for milk production/yield and quality. Contextually, these include the Ribosomal Protein L3 gene (*RPL3*), which has been described to have an especially high expression in the mammary gland, being involved in the regulation of energetic balance [122], or the Osteopontin gene (*OPN*), which encodes for a protein involved in many processes, such as cellular adhesion, chemotaxis and inflammation, cellular survival, and tissue remodelling, but is also potentially involved in milk production [123]. Furthermore, the Growth Hormone 1 gene (*GH1*) has been reported to likely be useful in stimulating the udder development in transgenic goats and, therefore, increasing their milk yield [48] as it has been reported for cattle milk production [124]. Sixth, the regulation of the expression of traits linked to hygienical quality of milk, measured through somatic cells counts, may be associated with the Lactoferrin gene (*LTF*), which is a very likely candidate gene for mastitis resistance. This occurs because, among the several functions in which it is involved, it has been reported to have a light antimicrobial effect in the mammary gland, which prevents the release of inflammatory mediators, such as the interleukins 1β and 6 or the α-tumoral necrosis factor [125]. This has also been reported for the Breast Cancer Type 1 gene (*BRCA1*), which encodes for a protein that is considered to be a DNA disorder-fixing molecule and to regulate the cell division cycle, as a potential early marker of disease, which has also been reported to increase somatic cell count in case of mastitis [126].

#### 3.2.2. Actual and Potential Candidate Genes Regulating the Expression of Milk Content/Composition

First, casein genes conform to the most widely explored milk trait genes’ expression regulation complex in the research. They attract special attention due to their implications for milk yield and quality determination [127]. Recent research advances in goat breeds such as Murciano-Granadina, Norwegian and Sarda goats have addressed the implication of the whole casein complex in milk production and contents (protein, fat, dry matter, lactose and somatic cells counts) [128,129,130,131,132,133,134,135,136,137,138,139]. The *CSN1S1*, *CSN1S2* and *CSN2* genes encode the αs1-casein, αs2-casein and β-casein, which are calcium-sensible caseins, while the *CSN3* gene encodes for κ-casein, which is involved in micelle stability [140]. Second, the α-Lactalbumin gene (*α-LA*) encodes for α-Lactalbumin, a highly important milk serum protein, which regulates the production of lactose in the milk of in all mammals. Contextually, this gene has been reported to increase lactose percentage in early-lactation milk from transgenics animals [141]. Third, the β-Lactoglobulin gene (*β-LG*) encodes for the β-Lactoglobulin, another essential milk serum protein, whose primary biologic functions seems to be related to phosphorus metabolism at the mammary gland, although its association with milk production and composition has not been comprehensively described in goat breeds [142]. Fourth, besides its acknowledged membrane drug and xenobiotics’ transporting activity, the ATP Binding Cassette Subfamily G Member 2 gen (*ABCG2*), is presumably linked to a decrease in milk production and an increase in milk protein and fat [118]. This correlated action in genes, regulating the expression between milk traits and those of another nature, has been relatively frequently addressed in the literature. For instance, the Insulin-like Growth Factor 1 gene (*IGF-1*) is associated, as mentioned in the previous sections, with body growth and the development or metabolism, but has also been reported to participate in regulating the expression of milk protein and fat, respectively [75].

Even in genes specifically reported to participate in the regulation of the expression of milk fat contents, a multifunction character has often been suggested. For instance, the Diacylglycerol Acyltransferase gene (*DGAT1*) encodes for a acetyl-coenzyme A variant, which is involved in lipid and other biomolecules’ synthesis, and was proved to associate with milk fat and protein contents [143]. The 1-acylglicerol-3-phosohate-O-acyltransferase gene (*AGPAT6*), which encodes for a crucial enzyme in glycolipids and triglycerides synthesis, the two main types of lipids present in milk, has been reported to present a high expression in the mammary gland epithelium; therefore, it has been suggested to have repercussions for milk fat percentage [144]. This was also noted for other genes, such as the Butyrophilin Subfamily 1 Member A1 gene (*BTNA1*), which was identified in a copy number variation analysis [84], and presents a widely acknowledged role in fat drops’ secretion in the mammary gland [145]. The Fatty Acid Synthase gene (*FASN*), which encodes for a protein involved in fatty acid synthesis, and the Acetyl-CoA Carboxylase gene (*ACACA*), which encodes for a liver fatty acids synthesis mediator protein, have been proved to present a certain association with milk lipid profile and fat percentage in dairy cattle, respectively [146,147]. The Stearoyl-CoA Desaturase gene (*SCD*) which encodes for a protein with a strong association to monounsaturated fatty acids synthesis in adipose tissue and mammary gland, has a direct repercussion for milk fatty acid profile [148]. The action of particular genes may not be direct but derive from these genes’ regulatory potential. Accordingly, the Peroxisome Proliferator-activated Receptor Gamma gene (*PPARγ*) encodes for a protein with a high impact on other lipid-metabolism-related genes’ transcription, such as *LPL*, *FASN*, *ACACA* y *SCD*, *PLIN2*, *PLIN3*, *FABP3*, *PNPLA2*, in same way as it regulates the expression of *NR1H3* y *SREBF1* genes [149]. The Adipophilin gene, also known as Perilipin 2 gene (*PLIN2*) [150] and the Perilipin 3 gene (PLIN3 ó TIP47) [151] take part in cytoplasmatic lipid transport and storage, and their implication for lipid secretion into milk has been widely reported. This also occurs at RNA levels, as was addressed for the Sterol Regulatory Element-binding Protein 1 (*SREBF1* or *SREBP-1c*). This encodes for a transcription factor *molecule*, which is a protein that regulates the expression of protein synthesis from mRNA, modulating the expression of those genes that encode for proteins with a fat milk percentage impact [152].

As those genes whose effects are highly expressed in the mammary gland should always be presumed to have an association with the expression of milk-related traits, either quantitative or qualitative. For example, the Patatin-like Phospholipase Domain Containing 2 gene (*PNPLA2*) encodes for a lipase that seems to be involved in intracellular triglycerides’ degradation in adipocytes, and for which overexpression in cattle mammary gland was proved [149], or the Aldehyde Dehydrogenase 2 gene (*ALDH2*), which is thought to take part in triglyceride synthesis and mammary gland epithelium cell apoptosis in bovine, with repercussions for fat milk yield and somatic cell count [153]. Many of the genes involved in fatty acid metabolism or fat mobilization have also been reported to participate in the regulation of the expression of milk quality traits, especially those reported with the fat fraction of milk. he faTtty Acid Binding Protein 3 gene (*FABP3*), which encodes for a protein participating in long-chain fatty acid intracellular transport, as well as other lipids’ cytoplasmatic movements, was proven to be associated with milk fat percentage [154]. Additionally, the Liver X Receptor-α (*NR1H3*) encodes for a transcription factor as well, ruling out the expression of other genes expression involved in fatty acids, cholesterol and carbohydrate metabolism, and was proved to contribute to milk lipid profile [155], as suggested for other genes, such as the Oxidized Low-Density Lipoprotein Receptor gene (*OLR1*), which encodes for a protein with fatty acid transport and oxidated-low density lipoprotein degradation functions, and has been reported to impact milk fat profile [156]. These genes not only directly affect the particular glands in which milk is produced, but may also be involved in the regulation of higher neurological routes. For example, the Brain-derived Neurotrophy Factor gene (*BDNF*), which encodes for a protein that regulates voluntary food intake and energetic distribution at the hypothalamic level, has been reported to be associated with fat milk percentage in cattle [157] or the Fat Mass and Obesity-associated Protein gene (*FTO*), which plays a central role in energetic homeostasis and waste control. This is, therefore, thought to affect the fat milk percentage [158]. The relationship across milk constituents may also be based on a correlated regulation of the function of certain genes, but this may affect other economically relevant traits, such as prolificacy. Therefore, the Acetyl-CoA Acetyltransferase gene (*ACAT*), which encodes for a cholesterol-metabolism regulating protein [159], has shown associations with other desirable productive traits (milk protein contents and fertility) in Holstein cattle [160] or the Long-chain Acyl-CoA Synthetase Isoform 1 gene (*ACSL1*), which encodes for a basic protein in triglycerides, phospholipids and cholesterol synthesis, making it a great candidate gene for milk-quality-related traits [161]. Table 5 presents a summary of the candidate genes associated with milk content that have been described in goat breeds, their chromosome location, exon counts, physiological function and where to find them in the literature, and Figure 4 presents a schematic representation of dairy goat candidate genes and their functions.

## 4. Conclusions

The knowledge concerning the main candidate genes affecting caprine meat and milk’s qualitative and quantitative production offers new opportunities to direct breeding practices towards the accurate and efficient selection of desirable traits. In this way, the latest genomic advances may allow to increase the response to selection, enabling a further genetic progress. There are pieces of evidence that suggest that the regulation of the expressions of very diverse traits (from body growth to milk composition) may somehow intertwine. This leads to the conclusion that gene regulation of the expression of caprine milk or meat very likely occurs in a multidimensional manner, taking place at different levels from central neurological control to the specific parts of the body, where, for instance, the meat and milk nutrients are produced. Consequently, knowing these functional regions of genomes may not only boost the efficiency, accuracy and progress of breeding schemes, but could also permit the reinforcement of local breeds’ conservation strategies by enhancing the sustainability and profitability of their products.

## Figures and Tables

**Figure 1 animals-12-00988-f001:**
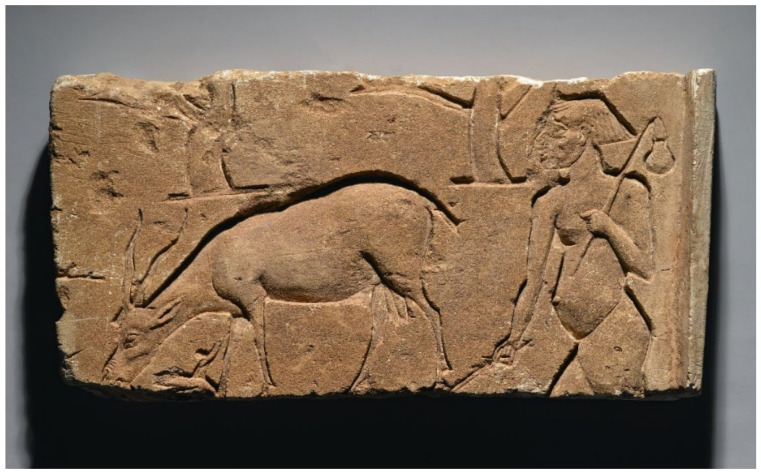
Relief Representation of Goatherd with Goat and Trees, ca. 1350–1333 BCE. New Kingdom, Amarna Period. Late Dynasty 18. Limestone, 8 1/4 × 16 3/4 × 2 1/2 in., 22.5 lb. (21 × 42.5 × 6.4 cm, 10.21 kg). Brooklyn Museum, Gift of the Ernest Erickson Foundation, Inc., New York, NY, USA, 86.226.30.

**Figure 3 animals-12-00988-f003:**
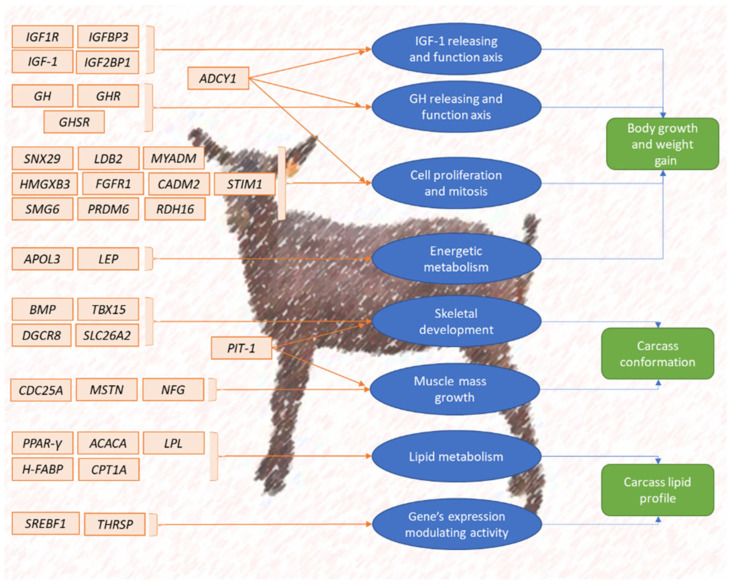
Summary scheme of caprine candidate genes’ influence on meat production.

**Figure 4 animals-12-00988-f004:**
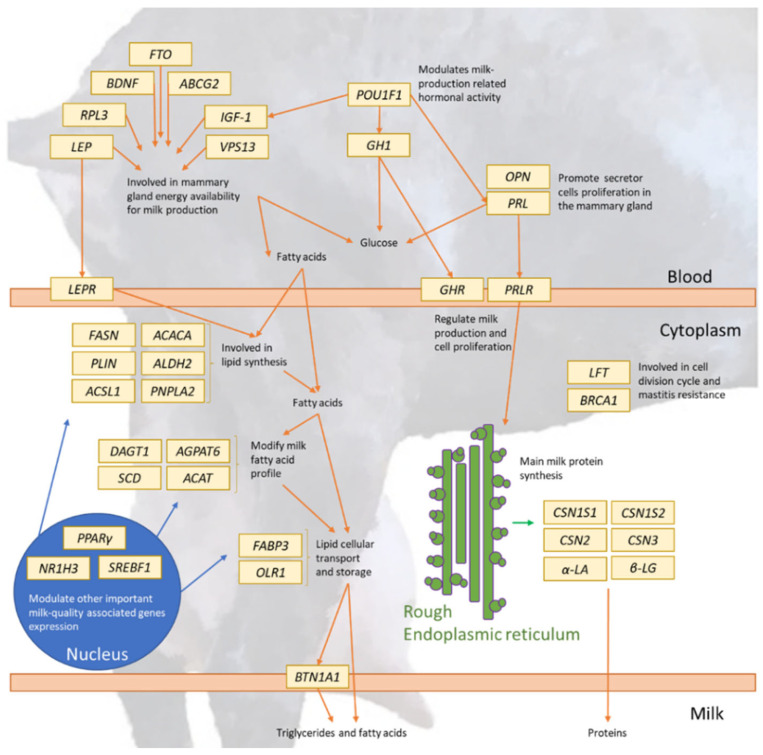
Dairy goat candidate genes interrelations’ scheme.

**Table 1 animals-12-00988-t001:** Breeding criteria and traits for caprine meat production.

Breeding Criteria	Traits	Reference
Body weight	Birth weight, body weight at 7, 14, 21 and 28 days, monthly weighing until 18 months of age	[56,57]
Growth	Average daily gain (ADG) before weaning, from 3 to 6 months of age, from 6 to 12 months of age	[57]
Conformation/corporal structure	Body length, height at the withers, chest girth, shoulder width and pin-bone width	[58]
Carcass quality	Hot (immediately after slaughter) and cold (before chilling) carcass weight; weights of the head, skin, heart, and thoracic and abdominal viscera; carcass dressing percentage; leg length and carcass width; and carcass fat coverage and lean meat and bone yield percentages	[56,59]
Meat quality	Muscle pH measurements, colour, water retention capacity, nutritional composition (protein, fat, and collagen percentages), and fatty acid profile in samples of *longissimus thoracis et lumborum*, *triceps brachii* and *semimembranosus*	[56]

**Table 2 animals-12-00988-t002:** Summary of the candidate genes associated with body growth and weight gain that have been described in goat breeds, their chromosome location, exon counts, physiological function and where to find them in the literature.

Acronym	Gene Name	Chromosome	Exon Count	Transcripts	Physiological Function	Goat Breeds in Which the Gene Has Been Described	Reference
*ADCY1*	Adenylate Cyclase 1	4	20	2	Involved in energetic metabolism, cellular mitosis and GH releasing.	Cameroon, West African Dwarf, Small East African and Landim goat	[60]
*SNX29*	Sorting Nexin 29	25	21	2	Regulates myoblast differentiation and proliferation.		
*LDB2*	LIM Domain Binding Factor 2	6	9	7	Modulates transendothelial leucocyte migration.	Nanjiang yellow goat	[83]
*MYADM*	Myeloid-associated Differentiation Marker	18	1	1	Involved in cellular membrane formation in a wide variety of cellular lines.	Bamu wild goat, Khonj wild goat, Australian feral Rangeland goats, Boer goats and Australian cashmere goat	[84]
*IGF1R*	Insulin-like Growth Factor 1 Receptor	21	21	1	Regulates IGF-1 activity.		
*APOL3*	Apolipoprotein L3	5	4	6	Involved in lipid blood transport and metabolism.	Leizhou goat	[70]
*STIM1*	Stromal Interaction Molecule 1	15	14	4	Associated with body weight gain.		
*HMGXB3*	HMG-Box Containing 3	7	21	3	Demonstrated relationship with cellular proliferation in neoplasia.		
*GH*	Growth Hormone	19	5	1	Related to corporal development.	Thai Native, Anglo-Nubian, Boer and Saanen goat	[85]
*GHR*	Growth Hormone Receptor	20	13	6			
*IGFBP3*	Insulin-like Growth Factor Binding Protein 3	4	5	1			
*BMP4*	Bone Morphogenic Protein 4	10	6	5		Boer goat, Chinese Xuhuai white goat and Chinese Haimen goat	[85]
*BMP15*	Bone Morphogenic Protein 15	10	2	1		Jining Grey goat	[85]
*IGF-1*	Insulin-like Growth Factor	5	7	9	Involved in corporal metabolism, growth and development.	Malabari and Black Attappady goat	[86]
*IGF2BP1*	Insulin-like Growth Factor Binding Protein 1	19	15	2	Involved in endocrine routes associated with body growth and development.	Shaanbei White Cashmere goat	[87]
*FGFR1*	Fibroblastic Growth Factor Receptor 1	27	20	13			
*SMG6*	Nonsense-mediated mRNA Decay Factor	19	20	2	Genome stabilization functions		
*CADM2*	Cell Adhesion Molecule 2	1	11	3	Involved in cellular migration and proliferation.	White and Black Guizhou, Nubiann, Boer and Huai goat.	[82]
*GHSR*	Growth Hormone Secretagogue Receptor	1	3	2	Regulates GH realise	Boer, Xuhuai and Haimen goat	[73]

**Table 3 animals-12-00988-t003:** Summary of the candidate genes associated with body size and carcass quality that have been described in goat breeds, their chromosome location, exon counts, physiological function and where to find them in the literature.

Acronym	Gene Name	Chromosome	Exon Count	Transcripts	Physiological Function	Goat Breeds in Which the Gene Has Been Described	Reference
*TBX15*	T-box Transcription Factor 15	3	9	2	Related to body size and chondrocytes and mesenchymal cell precursors regulator.	Gizhou Small goat	[98]
*DGCR8*	Drosha and DiGeorge Syndrome Critical Region microprocessor complex subunit gene 8	17	16	1	Associated with body size and involved in the osteoclastic development and remodelling bone activity.		
*CDC25A*	Cell Division Cycle 25 Homolog A	22	15	1	Responsible for corporal development and involved in myoblast differentiation and G1 quiescence.		
*LEP*	Leptin	4	3	1	Related to corporal and muscle mass development.		
*MSTN*	Myostatin	2	3	1			
*PRDM6*	PR/SET Domain 6	7	8	1	Regulates skeletal development and body mass index.	Shaanbei White Cashmere goat	[88]
*NGF*	Nerve Growth Factor	3	3	2	Involved in muscle and nervous tissue development.	Black Attappady and Malabari goat	[89]
*SLC26A2*	Solute Carrier Family 26 Member 2	7	9	8	Involved in bone tissue development.	Leizhou goat	[70]
*POU1F1*	Pituitary-specific Positive Transcription Factor 1	1	6	2	Plays an important role in corporal metabolism, growth and corporal development.	Thai Native, Ango-nubian, Boer and Saanen goat	[85]

**Table 4 animals-12-00988-t004:** Summary of the candidate genes associated with carcass lipidic profile that have been described in goat breeds, their chromosome location, exon counts, physiological function and where to find them in the literature.

Acronym	Gene Name	Chromosome	Exon Count	Transcripts	Physiological Function	Goat Breeds in Which the Gene Has Been Described	Reference
*PPAR-γ*	Peroxisome Proliferator-activated Receptors Gamma	22	6	1	Regulates adipose differentiation and lipid metabolism.	White Yaoshan goat	[59]
*LPL*	Lipoprotein Lipase	8	10	2	Involved in triglycerides degradation, obtaining glycerol and fatty acids.		
*H-FABP (FABP3)*	Heart Fatty Acid-Binding Protein (fatty acid binding protein 3)	2	5	2	Related to lipid metabolism.		
*SREBF1*	Sterol Regulatory Element-binding Protein 1	19	8	1	Modulates expression of other lipid-metabolism-related genes.		
*ACACA*	Acetyl-Coenzyme A Carboxylase	19	60	7	Involved in liver fatty acid synthesis.		
*THRSP* *(SPOT14)*	Thyroid Hormone Responsive	29	2	1	Regulates expression of other lipid-metabolism-related genes.		
*CPT1A*	Carnitine Palmitoyl-transferase 1A	29	19	1	Responsible for acyl-carnitine obtention, product of fatty acid degradation.	Moroccan goat breeds	[109]
*RDH16*	Retinol deshidrogenase-16	5	4	1	Involved in energetic metabolism.	Gizhou goat	[98]

**Table 5 animals-12-00988-t005:** Summary of the candidate genes associated with milk production/yield that have been described in goat breeds, their chromosome location, exon counts, physiological function and where to find them in the literature.

Acronym	Gene Name	Chromosome	Exon Count	Transcripts	Physiological Function	Goat Breeds in Which the Gene Has Been Described	Reference
*LEP*	Leptin	4	3	1	Regulates glycaemia, milk production and milk fat percentage.	Dairy cattle	[117]
*LEPR*	Leptin receptor	3	22	3			
*BDNF*	Brain-derived Neurotrophy Factor	15	6	5	Plays different roles in daily food intake and, in consequence, in nutrient and energy availability in the mammary gland.	Dairy cattle	[162]
*FTO*	Fat Mass and Obesity-associated Protein	18	9	1			
*IGF-1*	Insulin-like Growth Factor 1	5	7	9			
*ABCG2*	ATP Binding Cassete Subfamily G Member 2	6	22	8	Related to milk production and milk fat percentage.	Dairy cattle	[163]
*GHR*	Growth Hormone Receptor	20	13	6	Regulates cell growth, proliferation and apoptosis.	Dairy cattle	[117]
*PRLR*	Prolactin Receptor	20	11	7			[164]
*PRL*	Prolactin	23	5	1	Involved in mammary gland tissue preparation before lactation and regulates milk production.	Alpine goat	[165]
*RPL3*	Ribosomal Protein L3	5	10	1	Modulates energetic balance during the lactation yield peak.	Saanen goat	[70]
*VPS13*	Vacuolar Protein Sorting 13	8	73	3	Gene family associated with milk production.		
*VPS13B*	Vacuolar Protein Sorting 13 Homolog B	14	65	10	Related to leg’s morphological traits associated with fertility and milk production.	Dairy cattle	[115]
*VPS13C*	Vacuolar Protein Sorting 13 Homolog C	10	85	1	Regulates glycaemia increasing milk production.	Dairy cattle	[114]
*SPP1 (OPN)*	Osteopontin	6	7	2	Involved in milk yield production and milk fat percentage.	Dairy cattle	[123]
*GH1*	Growth Hormone 1	19	5	1	Seems to stimulate udder development and is associated with milk dairy traits.	Dairy cattle	[124]
*LTF*	Lactoferrin	22	17	1	Associated with mastitis resistance somatic cell count.	Damascus goat	[166]
*BRCA1*	Breast Cancer Type 1	19	23	5			
*POU1F1* *(PIT1)*	Pituitary-specific Positive Transcription Factor 1 (POU class 1 homeobox 1)	1	6	2	Modulates many milk-related hormones’ action.		
*CSN1S1*	αs1-casein	6	19	8	Associated with total milk protein composition and milk yield production.	Murciano-Granadina and Norwegian goats	[128,129,130,131,132,133,134,135,136,137,138,167]
*CSN1S2*	αs2-casein	6	19	7	Encodes for most of the important milk proteins.	Sarda goat	[128,129,130,131,132,133,134,135,136,137,138,167]
*CSN2*	β-Casein	6	9	2			
*CSN3*	Ƙ-Casein	6	5	1		Norwegian, Saanen, Canaria, Malagueña, Murciano-Granadina and Payoya goat	
*αLA LALBA (ALA)*	α-lactalbumin	5	4	1	Encodes for most of the important milk serum proteins.	White Inner Mongolia Cashmere, Xinong, Guanzhong, Laoshan, Leizhou, White and Black Guizhou and Banjiao, Matou goat	[168]
*β-LG* *LGB (PAEP)*	β-lactoglobulin/progestagen-associated endometrial protein	11	8	4		Zarayby, Damascus, Albino and Balady Hybrid goat	[142]
*DGAT1*	Diacylglycerol Acyltransferase	14	18	2	Involved in triglycerides synthesis.	Xinong, Saanen and Guanzhong goat	[169]
*AGPAT6* *(AGPAT6)*	1-acylglicerol-3-phosohate-O-acyltransferase	27	14	2			[144]
*BTN1A1*	Butyrophilin Subfamily 1 Member A1	23	9	5	Essential in milk lipid micelles secretion in the mammary gland.	Bamu wild goat, Khonj wild goat, Australian feral rangeland goat, Boer and Australian Cashmere goat	[84]
*FASN*	Fatty Acid Synthase	19	42	1	Plays an important role in fatty acids synthesis.		[84]
*ACACA*	Acetyl-CoA Carboxylase	19	60	7	Involved in liver fatty acid synthesis.	Saanen and ‘Grey local’ goat	[147]
*SCD*	Stearoyl-CoA Desaturase	26	6	1	Catalyze unsaturated to mono-satured fatty acids transformation reaction.	Boer, Xuhuai White and Haimen goat	[170]
*PPARγ* *(PPARG)*	Peroxisome Proliferator-activated Receptor Gamma	22	6	1	Regulates lipogenesis and adipogenesis.	Damascus goat	[166]
*PLIN3*	Perilipin 3	7	8	2	Involved in tissue fatty acid synthesis.	Not specified/Dairy goats	[149]
*PLIN2*	Perilipin 2	8	9	4			
*FABP3*	Fatty Acid Binding Protein 3	2	5	2	Involved in cytoplasmatic fatty acid storage and transport.		
*PNPLA2*	Patatin-like Phospholipase Domain Containing 2	29	11	3	Encodes for one of the proteins involved in fat micelles formation.		
*SREBF1*	Sterol Regulatory Element-binding Protein 1	19	8	1	Transcription factors involved in lipid homeostasis.		
*NR1H3*	Liver X Receptor-α (nuclear receptor subfamily 1 group H member 3)	15	10	2			
*OLR1*	Oxidized Low Density Lipoprotein Receptor	5	6	2	Involved in lipid transport and oxided-LDL form degradation.	Sirohi goat	[171]
*ALDH2*	Aldehyde Dehydrogenase 2	17	13	1	Associated with triglycerides synthesis in mammary glandular tissue.	Xinong Saanen goat	[172]
*ACAT1*	Acetyl-CoA Acetyltransferase 1	15	12	1	Involved in cholesterol metabolism.		
*ACAT2*	Acetyl-CoA Acetyltransferase 2	9	9	1			
*ACSL1*	Long-chain Acyl-CoA Synthetase Isoform 1	27	24	5	Participates in triglycerides, phospholipids and cholesterol synthesis.		
*BDNF*	Brain-derived Neurotrophy Factor	15	6	5	Plays different roles in daily food intake and, in consequence, in nutrient and energy availability in the mammary gland.	Dairy cattle	[162]
*FTO*	Fat Mass and Obesity-associated Protein	18	9	1			
*ABCG2*	ATP Binding Cassete Subfamily G Member 2	6	22	8	Related to milk production and milk fat percentage.	Dairy cattle	[163]

## Data Availability

Not applicable.

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
