# Peer review of "Candidate Genes and Their Expressions Involved in the Regulation of Milk and Meat Production and Quality in Goats (Capra hircus)"

_animals, 2022, doi:10.3390/ani12080988_

Round 1
Reviewer 1 Report
The objective of this review was to make a brief chronological analysis of the advances made in caprine genetics and genomics and to describe the main candidate genes for caprine meat and milk production and quality, which have been reported in literature up to the date.
This is a very good and well-written review manuscript. I only suggest considering next minor comments to improve the manuscript:
- Lines 35-37: Please consider in the abstract “the chronological analysis in caprine genetics and genomics” as part of the aim of this manuscript.
- Line 138: Please make sure that Figure 2 appears in the text.
- Line 210: I suggest replacing “determine” by “describe” or “summarize”.
- Line 252: Replace “characterizes” by “is characterized”.
- Line 254: Replace “Improving” by “improving”.
- Lines 277-278: This description corresponds only to Table 1, but not to Figure 3. Please make sure to include the description for Figure 3 in the right place in the text.
- Lines 367-373: Very long paragraph, please separate it in two shorter paragraphs.
- Line 576: The reference number 155 is not in order of appearance.
- Page 610: For some genes the information presented in Table 5 is not well aligned making it difficult to read. Correct the size of the reference numbers. The “IGF-1” gene is repeated as it appeared in both pages 19 and 20.
- Page 668: Remove the space between name initials and the end of the line.
- Page 691: It appears that page range is missing.
- Line 699: Correct journal abbreviation.
- Line 721: Correct author name (i.e., A.-L.).
- Line 774: Correct author name (i.e., M.-M.).
- Line 783: Replace “one” by “One”.
- Line 795: Journal name, volume and page range are missing.
- Line 796: Correct author’s names.
- Line 817: Page range is missing.
- Line 835: Journal name should be italic.
- Line 852: Correct author’s names.
- Line 856: Correct author’s names.
- Line 862: Replace “one” by “One”.
- Line 865: Correct author’s names.
- Line 909: Page range is missing.
- Line 968: Page range is missing.
- Line 993: Journal year and page range are missing.
- Line 1019: Correct author’s names.
Author Response
Reviewer 1
Comments and Suggestions for Authors
The objective of this review was to make a brief chronological analysis of the advances made in caprine genetics and genomics and to describe the main candidate genes for caprine meat and milk production and quality, which have been reported in literature up to the date.
This is a very good and well-written review manuscript. I only suggest considering next minor comments to improve the manuscript:
Response: We thank the reviewer for his/her kind comments.
- Lines 35-37: Please consider in the abstract “the chronological analysis in caprine genetics and genomics” as part of the aim of this manuscript.
Response: Added.
- Line 138: Please make sure that Figure 2 appears in the text.
Response: A reference to Figure 2 was added.
- Line 210: I suggest replacing “determine” by “describe” or “summarize”.
Response: We followed the reviewer’s suggestion.
- Line 252: Replace “characterizes” by “is characterized”.
Response: We followed the reviewer’s suggestion.
- Line 254: Replace “Improving” by “improving”.
Response: Replaced.
- Lines 277-278: This description corresponds only to Table 1, but not to Figure 3. Please make sure to include the description for Figure 3 in the right place in the text.
Response. We followed the reviewer suggestion, added a reference to figure 3 in the proper place and changed Figure 3 to a position in which it may be more relevantly placed.
- Lines 367-373: Very long paragraph, please separate it in two shorter paragraphs.
Response. We followed the reviewer suggestion.
- Line 576: The reference number 155 is not in order of appearance.
Response: Corrected.
- Page 610: For some genes the information presented in Table 5 is not well aligned making it difficult to read. Correct the size of the reference numbers. The “IGF-1” gene is repeated as it appeared in both pages 19 and 20.
Response. We followed the reviewer suggestion.
- Page 668: Remove the space between name initials and the end of the line.
- Page 691: It appears that page range is missing.
- Line 699: Correct journal abbreviation.
- Line 721: Correct author name (i.e., A.-L.).
- Line 774: Correct author name (i.e., M.-M.).
- Line 783: Replace “one” by “One”.
Response: Changed.
- Line 795: Journal name, volume and page range are missing.
Response: Added.
- Line 796: Correct author’s names.
Response: Corrected.
- Line 817: Page range is missing.
Response: Corrected.
- Line 835: Journal name should be italic.
Response: Corrected.
- Line 852: Correct author’s names.
Response: Corrected.
- Line 856: Correct author’s names.
Response: Corrected.
- Line 862: Replace “one” by “One”.
Response: Changed.
- Line 865: Correct author’s names.
Response: Corrected.
- Line 909: Page range is missing.
Response: Added.
- Line 968: Page range is missing.
Response: Added.
- Line 993: Journal year and page range are missing.
Response: Added.
- Line 1019: Correct author’s names.
Response: Corrected.
Reviewer 2 Report
This is an interesting review regarding to goat production, particularly with a longitudinal overview along the studies in history.
Major concerns
- The title is suggested as “Candidate genes and their expressions involved in the regulation of the expression of milk and meat production and quality in goats (Capra hircus)”.
- The Simple summary is too long and tedious, the writings looks like an introduction.
- The authors need to further delineate the passage by the genes listed in the text to affect milk and meat production, namely through the abundance of their transcripts or by the sequence variations such as point mutation, deletion, insertion, or even in association with SNP or other markers.
Author Response
Reviewer 2
Comments and Suggestions for Authors
This is an interesting review regarding to goat production, particularly with a longitudinal overview along the studies in history.
Response: We thank the reviewer for his/her kind comments.
Major concerns
- The title is suggested as “Candidate genes and their expressions involved in the regulation of the expression of milk and meat production and quality in goats (Capra hircus)”.
Response: We followed the reviewer suggestion but removed the second time that the word expression was used to avoid redundancies.
- The Simple summary is too long and tedious, the writings looks like an introduction.
Response: Simple summary was reduced to almost half its former extension.
- The authors need to further delineate the passage by the genes listed in the text to affect milk and meat production, namely through the abundance of their transcripts or by the sequence variations such as point mutation, deletion, insertion, or even in association with SNP or other markers.
Response: We followed the reviewer suggestion and added the number of transcripts per referenced gene for each of the genes considered in this paper.